# Effects of Complex Pain Control Programs on Taekwondo Athletes with Recurrent Low Back Pain: A Case Study

**DOI:** 10.3390/medicina59071271

**Published:** 2023-07-08

**Authors:** Hong-Gil Kim, Ju-Hyeon Jung, Dong-Chul Moon

**Affiliations:** 1Department of Physical Therapy, Graduate School, Dong-Eui University, Busan 47340, Republic of Korea; rlaghdrlf456@gmail.com; 2Department of Physical Therapy, College of Nursing, Healthcare Sciences and Human Ecology, Dong-Eui University, Busan 47340, Republic of Korea; 3Department of Physical Therapy, Gimhae College, Gimhae-si 50811, Republic of Korea; ptmdc@hotmail.com

**Keywords:** recurrent low back pain, taekwondo, mechanosensitivity

## Abstract

*Background and Objectives*: Practitioners of martial arts such as Taekwondo are likelier to experience back pain during training or competition. As the back pain of taekwondo athletes shows various symptoms depending on the athlete’s characteristics, such as technique and movement, a case study was conducted to verify the intervention effect suitable for individual traits. We examined the effects of a complex pain control program on pain, mechanosensitivity, and physical function in a Taekwondo athlete with recurrent low back pain (LBP). *Materials and Methods*: A Taekwondo athlete with LBP was recruited from D University, Busan. The intervention program was performed for 45 min twice a week for 3 weeks, and the patient was followed up with after 2 weeks. The numerical rating pain scale (NRPS), pain pressure threshold, mechanosensitivity, and Oswestry Disability Index (ODI) scores were measured before and after the intervention. Therapeutic massage and nerve stimulation therapy were performed. Lumbar flexion, extension, and rotation were performed in the movement control exercise group, whereas the sliding technique, a neurodynamic technique of the tibial nerve, was applied in the neurodynamic technique group. This effect was verified by comparing the average measured values before and after the intervention. *Results*: Pain (NRPS) and mechanosensitivity reduced, range of motion and tactile discrimination abilities improved, and physical function (ODI) improved. The effect of the improved intervention lasted 2 weeks. *Conclusions:* These results indicate that application of complex pain control programs considering the four aspects of pain mechanisms for 3 weeks can be an effective intervention in Taekwondo athletes with recurrent LBP.

## 1. Introduction

With the recent increase in the intensity and frequency of sports participation in the general population, the incidence of low back pain (LBP) has continuously increased in modern society [1]. Under the complex influence of physical and psychological factors such as lifestyle, anxiety, stress, and depression, LBP increases in duration and repeating intervals of pain as it develops into chronic or recurrent pain [2]. Recently, the International Association for the Study of Pain has reported that various types of pain, including LBP, are caused by the complexity of five different mechanisms (nociplastic, neuropathic, nociceptive, motor control, and psychological) [3]. Taekwondo practice includes both physical and mental aspects. The physical part is challenged to increase muscle strength, flexibility, speed, and skill through various training exercises and techniques. Taekwondo players continuously practice formality (patterns of movement) and sparring. Through this, Taekwondo athletes experience back pain as frequently as the general population [4,5]. LBP has been reported as a common pain in Taekwondo [6]. Studies have shown that the back is the third most injured body part in elite adult Canadian Taekwondo athletes [7]. Although there are known data that LBP frequently occurs in judo and karate athletes, studies focusing on frequent LBP in Taekwondo athletes are lacking [8].

Recurrent LBP (RLBP) frequently occurs among athletes performing martial arts sports, including Taekwondo, who predominantly perform high-intensity and repetitive training involving sparring with partners, an activity that induces various injuries. The time spent on recovery is short, and repeated overloading is applied to the lumbar spine [9]. As such, LBP affects athletic performance [10]. Athletes with RLBP often show the following symptoms: synesthesia in the area of pain, reduced range of motion (ROM), nerve signs, and radiating pain [11], as well as compensatory responses to avoid pain, which can cause movement control disabilities [12].

Athletes commonly spend considerable time and money on surgery and rehabilitation therapies to control the symptoms of RLBP [13]. Previous studies have reported that athletes with LBP may undergo a general physical therapy intervention that has long been used for pain control or that they may need to take time off from training to facilitate recovery. Other applied methods include ultrasound, kinesio-taping, transcutaneous electrical nerve stimulation, and manual therapy [14]. However, recent studies on RLBP rehabilitation have shown that, despite the short-term positive effects, general physical therapy interventions have negligible effects on long-term management [15]. A previous study indicated that the lack of long-term effects of general therapeutic interventions were because they were based on a single rather than on several pain mechanisms that could occur simultaneously in athletes [16]. This has led to recent claims that the effectiveness of the intervention may be maximized through a pain-mechanism-based therapy approach, reflecting several simultaneous pain mechanisms in athletes with RLBP [17,18,19,20]. Medical therapies commonly apply such mechanism-based approaches [21,22], which have several advantages, including a widened scope of therapeutic intervention for physical therapists, and the possibility of reflecting the results of the latest studies in various fields [3]. Nevertheless, relatively few studies have investigated mechanism-based therapies. This study therefore aimed to verify the effects of complex pain control programs incorporating mechanism-based therapeutic approaches in Taekwondo athletes with LBP.

## 2. Materials and Methods

### 2.1. Patient Characteristics

The current case study involved subjects who had experienced repetitive back pain at least twice in the past six months due to high-intensity training involving repetitive kicking movements for more than 5 h, five days a week, accompanied by pain.

He is a 23-year-old subject with RLBP from D University in Korea and a registered athlete with the Korea Taekwondo Athletes Association with more than ten years of experience. He was diagnosed with L5~S1 disc spondylosis through MRI at a general hospital in Busan in July 2020. The subject’s Visual Analogue Scale (VAS) score was 5 points, and his ODI (Oswestry Disability Index) score was 8 points, which belonged to the selection criteria of previous studies [17]. In addition, subjects are those who complain of pain due to visceral diseases such as cancer and heart disease, those who have a risk of increasing the intensity of pain during intervention due to severe pain, those who have surgery, scoliosis, neurological symptoms, and skin conditions. The experiment did not use people who cannot measure and intervene due to hypersensitivity [17].

The subject used a motion analysis device (MyoMotion Clinical, Noraxon, Scottdale, USA Inc.) to attach sensors to S1 and T12 and conduct motion analysis. He had lumbar spine LOM due to radiating pain. In addition, radiating pain was observed in the left leg during the SLR test while maintaining the lumbar curve in the supine position. The subject was diagnosed with back pain in the hospital and did not receive any medications or treatment until the trial intervention was conducted.

The evaluation and intervention in this study were conducted by different physical therapists with more than six years of clinical experience. This study was approved by the Institutional Review Board of Dong-Eui University (DIRB-202202-HR-R-02). Subjects received an adequate description of the study purpose and intervention (Table 1).

### 2.2. Procedures

Before the intervention, tactile discrimination (TD) and mechanosensitivity (VAS, muscle activation [MA], ROM, and ODI) were measured in the given order. Measurements were repeated every week during the 3 weeks of intervention and 2 weeks after the completion of the intervention to monitor the continuous effects via a re-test. Among the measured variables, TD was used to assess changes in skin sensation in the area of pain related to chronic lumbar pain. For the measurement, nine blocks were drawn on the patient’s back, where the patient selected three pieces of paper indicating numbers 1–9 that had been randomly arranged twice; the patient was placed in a prone position, where the TD was obtained as the center of each block was pressed. Triplicate measurements were performed, and the mean was calculated for the analysis (Figure 1).

Mechanosensitivity indicates the sensitivity of peripheral nerves to movement and was used in this study to assess the changes in nerve sensitivity induced by the intervention [23]. Lumbar and pelvic movements were limited in active SLR (ASLR), as the pressure of the biofeedback unit (PBU, Chattanooga, USA) was applied to an area below the navel of the patient, who was guided to maintain 40 mmHg of PBU pressure with an error range of 10 mmHg. The point at which the patient first sensed pain during ASLR was defined as P1, and the point at which he sensed the maximum pain to be unable to produce further changes in ROM was defined as P2. 

The VAS score and ROM at P1 and P2 were measured each week during the period of intervention, and before the intervention, the VAS score and MA were measured simultaneously on the ROM at P1 and P2. The ODI was used to assess pain-related discomfort during daily activities.

### 2.3. Measurement Methods

#### 2.3.1. Tactile Discrimination

In the measurement of TD, a total of nine squares, three along a 5-cm width and three along a 5-cm length, were drawn on the lumbar area based on two-point discrimination, where two points were sensed as a single point to define the size of one square. Each square was marked by numbers 1–9, and the rater guided the patient to be aware of the squares from 1 to 9 for 5 min and to select a random sequence of numbers twice for each number on the square. Using a pen to press the center of each square indicated by the selected sequence of numbers, the rater asked the patient to correctly guess the number of times the square was pressed. After all three sequences of numbers selected by the patient were tested, the rater converted the number of times the patient had correctly guessed into a score and estimated the mean [16,17,18,19,20,21,22] (Figure 2).

#### 2.3.2. Mechanosensitivity

##### Visual Analogue Scale

In the ASLR test, the point at which the patient first sensed pain was set as P1, and the point at which he sensed the maximum pain to be unable to produce further changes in the ROM was set as P2. The intensity of pain at P1 and P2 was expressed by the patient to the rater as a VAS score, and the mean of triplicate measurements was used for the analysis [23,24,25].

##### Muscle Activation

In the ASLR test, the patient was instructed to maintain his ankle at a 90° angle to prevent lumbar flexion and to measure the mechanosensitivity of the lower limb during pure hip flexion. In addition, PBU (Chattanooga, USA) was used to control the effect of lumbar flexion on the nerves going to the legs. The pressure of PBU was maintained at 40 mmHg. The margin of error for the pressure change during ASLR was set at 10 mmHg.

The electrodes for surface electromyography (TeleMyo Desktop DTS, Noraxon, Scottsdale, AZ, USA) for the measurement of MA were attached to the origin of the biceps femoris and semimembranosus muscles, as well as to the muscle belly at a two-third position between the part and insertion. The distance between the two electrodes was 2 cm. For the lateral gastrocnemius, the electrodes were attached to the muscle belly 2 cm below the knee joint [24]. The MA during ASLR was measured simultaneously with the VAS and ROM.

MyoResearch Master Edition 3.10 (Noraxon, Scottsdale, AZ, USA) was applied to analyze the MA data. The extraction speed for the surface electromyography signals was set at 1024 Hz, and the interference was removed using a low-pass filter at 20–350 Hz.

The root mean square (RMS) values measured for 3 s for each muscle (excluding the first and last seconds) were used as variables. For MA, triplicate measurements were performed at P1 and P2, and the respective means were used. Based on the initial ROM values at P1 and P2, the differences in the mechanical sensitivity of the biceps femoris, semimembranosus, and lateral gastrocnemius were measured each week. To normalize the measured values, the values of maximum voluntary isometric contraction (MVIC) for each muscle before the intervention were used to calculate the %MVIC [25] (Figure 3).

##### Range of Motion

During ASLR (Active straight leg raise), the rater measured the changes in hip joint ROM and ROM at P1 and P2 by attaching a motion analysis device (Myomotion, Noraxon, Scottsdale, AZ, USA) to L3, the center of both posterior superior iliac spines, a one-tow point at the leg, and a one-tow point at the tibia and center of the talus. The ankle joint angle was maintained at 90° to identify the pure angle of the hip joint. The VAS score, ROM, and MA were simultaneously measured [25,26].

#### 2.3.3. Oswestry Disability Index

The patients completed the questionnaire at onset and each week before the intervention. The questionnaire comprised the following 10 items investigating daily activities: pain intensity, personal care, lifting, walking, sitting, standing, sleeping, sex life, social life, and traveling. Each item is rated on a scale of 0–5, and the patient could respond yes or no to the four questions of each item regarding their personal status in the past 24 h [27].

### 2.4. Intervention

The intervention in this study to test the complex pain programs based on four aspects of pain mechanisms comprised three sets of 5-min TD training, 10-min neurodynamic exercise, 15-min movement control exercise, and 10-min therapeutic massage and stretching. The total reaction time was 40 min. The intervention was administered twice weekly for three weeks, considering the subject’s training and competition schedule [28]. The patients received explanations regarding the intervention before participating in the therapy (Figure 3).

#### 2.4.1. Tactile Discrimination Training

On the patient’s back, the rater prepared a total of nine blocks, three each with a 5 cm width and length. Each block was assigned a number from 1 to 9. The rater then pressed the center of the block with a pen and showed the patient an image of the block numbered 1–9. The goal is to correctly guess the pressed block. The back of the pen was used to press the numbered blocks, and the patient was educated for being familiarized with each of the nine numbered blocks. During this period of 5 min of training, the patient was repeatedly informed of the correct number of blocks being pressed with repeated stimulation on the block in the case of guessing an incorrect number until he was correctly aware of all nine blocks [22] (Figure 2).

#### 2.4.2. Neurodynamic Exercise

The rater provided manual shoulder support to maintain pure cervical flexion. While maintaining extension of the toe and ankle, the patient performed knee extension until the leg muscles felt fully stretched or started to cramp. The patient performed neck extension immediately before the muscle was stretched or clamped for further knee extension. This was followed by cervical and knee flexion to complete three sets of sliding techniques, with each set comprising 10 exercises and 1 min of rest [18] (Figure 4).

#### 2.4.3. Movement Control Exercise

Training involved flexion, extension, and rotation. During training in the flexion direction, the patient was guided to maintain a neutral pelvic position in the quadrupedal position and to move the hip joint from 90° to 120°. During training in the extension direction, the subject was guided to maintain a neutral pelvic position up to 120° flexion of both knees, as the PBU (Chattanooga, USA) was used, and a pressure of 40 mmHg was maintained on the abdomen in the prone position. During training in the rotation direction, the patient was guided to maintain a neutral pelvic position in the side-lying position up to 15° abduction of the hip joint [29] (Figure 5).

#### 2.4.4. Therapeutic Massage and Stretching

To increase the flexibility of the tensed muscles, therapeutic massage was performed on the quadratus lumborum and erect spine muscles, followed by 10 min of hamstring stretching [30,31].

## 3. Results

Compared with the pre-intervention scores, the TD point (TDP) scores after the 3-week intervention increased by 21.43%, while scores at the 2-week follow-up increased by 33.33% (Table 2, Figure 6).

The VAS scores at P1 and P2 during ASLR decreased after the 3-week intervention compared to the initial scores by 40% at P1 and 20% at P2 after the 3-week intervention compared with the initial scores. The scores at the 2-week follow-up decreased by 48% at P1 and 34.29% at P2 compared with the initial scores, and the levels were subsequently maintained (Table 3, Figure 7).

The ROM scores at P1 and P2 during ASLR increased following the 3-week intervention by 16.79% at P1 and 13.01% at P2 compared with the initial scores. The scores during the 2-week follow-up after the intervention increased by 13.42% at P1 and 8.85% at P2 compared with the initial scores, and the levels were subsequently maintained (Table 4, Figure 8).

The electromyography activity scores of the semimembranosus, biceps femoris, and gastrocnemius muscles for the P1 and P2 ROMs in the early stage of ASLR decreased after the 3-week intervention compared with the initial scores by 22.37% at P1 and 48.82% at P2 in the semimembranosus muscle, 13.77% at P1 and 46.76% at P2 in the biceps femoris muscle, and 14.78% at P1 and 18.37% at P2 in the gastrocnemius muscle. The scores during the 2-week follow-up after the intervention decreased by 35.53% at P1 and 42.74% at P2 in the semimembranosus muscle, by 31.77% at P1 and 62.3% at P2 in the biceps femoris muscle, and by 29.14% at P1 and 38.42% at P2 in the gastrocnemius muscle, compared with the initial scores, and the levels were subsequently maintained in all three muscle groups (Table 5, Figure 9).

Regarding the ODI, the scores after the 3-week intervention were 25% lower than the initial scores, and the scores decreased by 37% compared with the monitor post-intervention levels during the 2-week follow-up (Table 6) (Figure 10).

## 4. Discussion

This study aimed to determine the effects of complex pain control programs involving the treatment of four of five pain mechanisms on TD, tissue mechanosensitivity, and lumbar functions in Taekwondo athletes with recurrent LBP.

The inclusion criteria for patients with RLBP in previous studies were individuals who complained of pain at least twice in the past 6 months, with a level of pain estimated as a VAS score ≥ 5 and an ODI score of ≥5, while the exclusion criteria were individuals with acute lumbar pain, spinal fracture, and a history of surgery, spondylolisthesis, or spinal tuberculosis [11]. The same criteria were used to determine the eligibility of the present patient.

Advances in neuroscience and brain imaging research have allowed further research in the human brain, which have reported that the brain undergoes functional changes when a part of the body experiences pain [32]. Pain-induced functional changes in the brain prevent normal cortical processing, causing complex cognitive, sensory, and motor control [33]. In a previous study, an intervention combining TD and motor control training had a positive effect in individuals with RLBP [16]. Similarly, in this study, the methods in the previous study were modified to apply a nine-point TD training during the 3-week intervention; the TDP scores were increased from 11-point pre-intervention to 16.5-point post-intervention, the latter of which was maintained during the 2-week follow-up period.

Previous studies have shown that motor control training leads to the control of compensation movement via education of the order of muscle recruitment, thereby reducing the load on the joint caused by the compensation movement [34,35]. Motor control training combined with TD training has been shown to induce changes in the cerebral cortex governing the lumbar pain area to correct paresthesia in the pain area, causing misinterpretation of normal stimuli from peripheral nerves [36]. These claims lend support to the trend of an increase in TDP following the intervention combining TD training and motor control training in this study.

Mechanosensitivity indicates the level of nerve tissue activity in response to a mechanical force [37]. A previous study reported that an increase in pain in patients with RLBP caused an increase in inflammation of the nerve and surrounding tissues, while decreasing nerve mobility and increasing pressure caused by edema around the nerve itself and circulatory disability [38]. This phenomenon could also increase the pain in patients to induce involuntary contraction of the muscles in the vicinity, thereby causing pain [38]. An increase in mechanosensitivity is indicative of a state in which even a small or normal stimulus is sensed as risk or pain [39]. A previous study verified the effects of a neurodynamic technique applied to peripheral nerves to reduce pain, to increase hip joint ROM, and to decrease the activity of lower limb muscles in performing specific movements [23]. Based on this study, the present investigation applied ASLR to test the reduction in mechanosensitivity and measure the subsequent pain, hamstring muscle activity, and hip joint ROM of the patient. The detailed methods used to measure these variables in this study were as follows: The point of onset of pain (P1) and point of maximum pain (P2) were set, and the scores were measured and compared pre-intervention, post-intervention after 3 weeks, and at follow-up after 2 weeks. To facilitate an accurate comparison of the pain levels at P1 and P2 at pre-intervention, post-intervention, and follow-up, the P1 and P2 scales at post-intervention and follow-up were compared based on the angles of P1 and P2 at pre-intervention. Meanwhile, the change in ROM was examined by recording the newly sensed angles of P1 and P2 after the 3-week intervention, using a motion analysis device. Simultaneously, the MA of the biceps femoris and semimembranosus muscles were measured to verify the reduction in mechanosensitivity in the lower limb, in parallel with the decrease in hamstring MA and increase in hip joint ROM.

The results of this study showed that an intervention combining neurodynamic, movement control, and TD training facilitated a reduction in pain at P1 and P2 when the patient performed SLR. The angles at P1 and P2 that were reassessed after the intervention both showed an increase, which was interpreted as an increase in ROM. In addition, MA was shown to decrease the angles of the hamstring and gastrocnemius muscles, and the altered level from the initial pre-intervention level was maintained for 2 weeks post-intervention. In a previous study, motor control exercises could discriminate between the normal alignment of the body and isolated motions at each segment to disperse the load on each segment [40]. Previous studies applying the neurodynamic technique reported that it could lower the hypersensitivity of the senses and tissues by reducing the pressure on the nerves and surrounding tissues and improving blood flow to the muscles and nerves [36,38,41,42,43]. As such, the intervention program applied in this study is presumed to have improved both the pain and function of the lumbar area, as the neutral spinal position was recognized through movement control training, while the nearby joints were controlled to prevent excessive motion such that mechanosensitivity and pain were reduced in the surrounding tissues and the nerve fibers could recover from inflammation and hypoxia. Additionally, the key factor in the intervention program in this study was the effect of TD training. Previous studies have reported that pain could cause the cerebral cortex to fail to accurately identify the normal information sent from the body segments and recommended TD training as a solution to improve such errors in the cerebral cortex [22]. The TDP results in this study indicated that the increases in scores after the intervention were due to the TD training effect. TD training, in which the lumbar area was divided into nine blocks for repeated practice of accurate identification of specific blocks, is likely to improve cognitive abilities regarding segments.

In several previous studies, the ODI was used to objectively measure pain in patients with LBP and to examine the functional levels associated with functional and cognitive impairments [44,45,46]. Previous studies have shown that the use of a complex intervention, including TD training, could induce positive changes in body functions in patients with lumbar pain [16,22]. Similarly to this study, the ODI scores increased after the intervention, while the increased levels were maintained during the 2 weeks following the intervention. Factors related to pain and daily activities increased among the ODI scores. This is presumed to be because the intervention program in this study ultimately improved the pain in the subjects, and the reduced pain could improve movements, which in turn improved physical function.

The findings of this study suggest that complex pain control programs are effective for the management of pain in Taekwondo athletes with LBP. Through the management and improvement of the four pain mechanisms, the pain-related reduction of mechanosensitivity in the surrounding tissues and body’s functions and abilities to discriminate the segments could be improved. Based on these results, it is predicted that the use of a complex intervention based on pain mechanisms rather than a single perspective, even in the early days of rehabilitation, would be effective at promoting the functional recovery of training athletes with RLBP.

This study had some limitations. First, by applying for the intervention program, only secondary functional changes were confirmed without confirming changes in the activity of the back muscle. Second, the effects of a single intervention program could not be confirmed by confirming the impact of a complex intervention program. Third, an approach that reflects psychological and social factors related to chronic pain through repeated trauma and injury in Taekwondo players has not been investigated through various pain mechanisms.

Therefore, further studies are needed to determine the long-term effects of complex pain control programs based on pain mechanisms, including psychological and social factors.

## 5. Conclusions

Complex pain control programs incorporating these four aspects of pain mechanisms can reduce pain and mechanosensitivity, improve ROM and TD, and enhance physical function. As such, we propose that such complex pain control programs could be effective interventions for Taekwondo athletes with LBP.

## Figures and Tables

**Figure 1 medicina-59-01271-f001:**
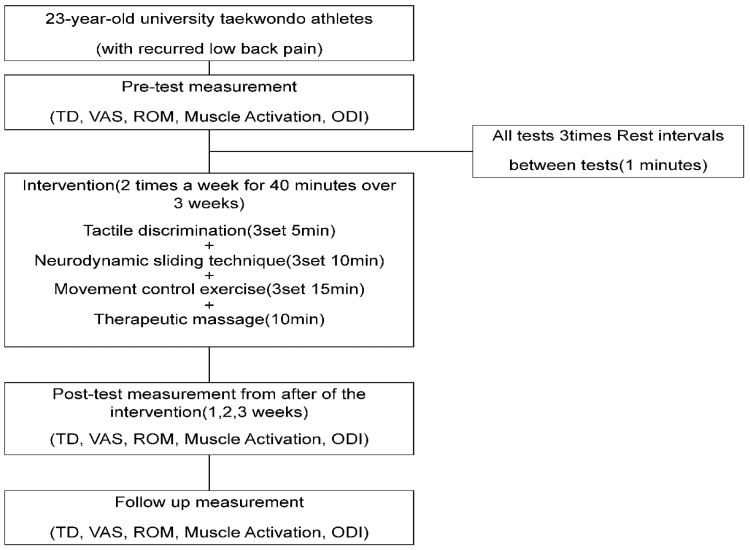
Flowchart of the study.

**Figure 2 medicina-59-01271-f002:**
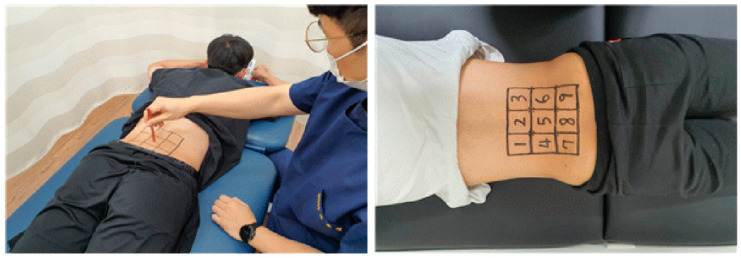
Tactile discrimination test and training.

**Figure 3 medicina-59-01271-f003:**
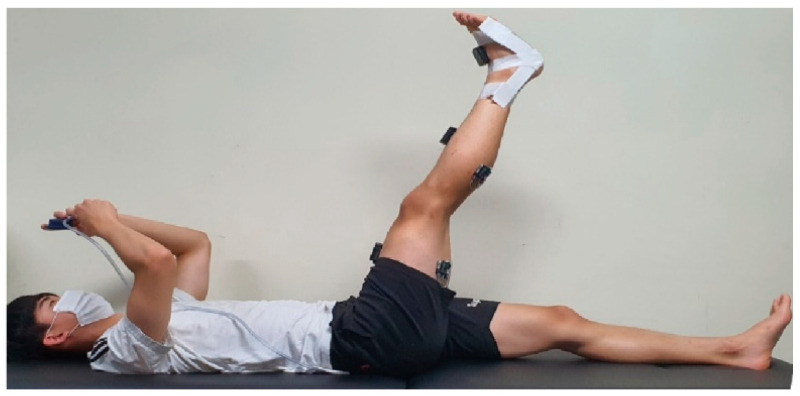
Electromyography measurement of the lower extremity during the ASLR test.

**Figure 4 medicina-59-01271-f004:**
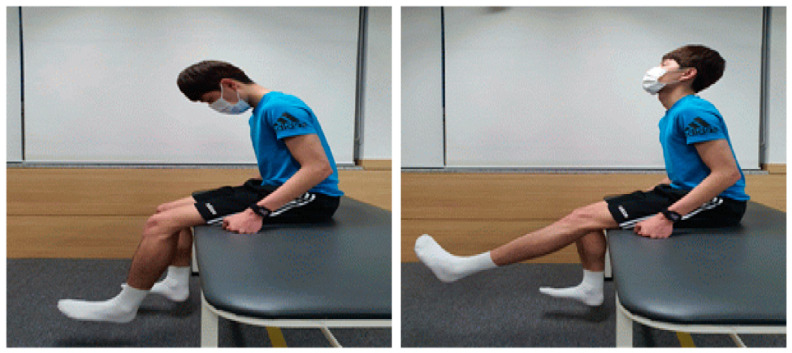
Neurodynamic training.

**Figure 5 medicina-59-01271-f005:**
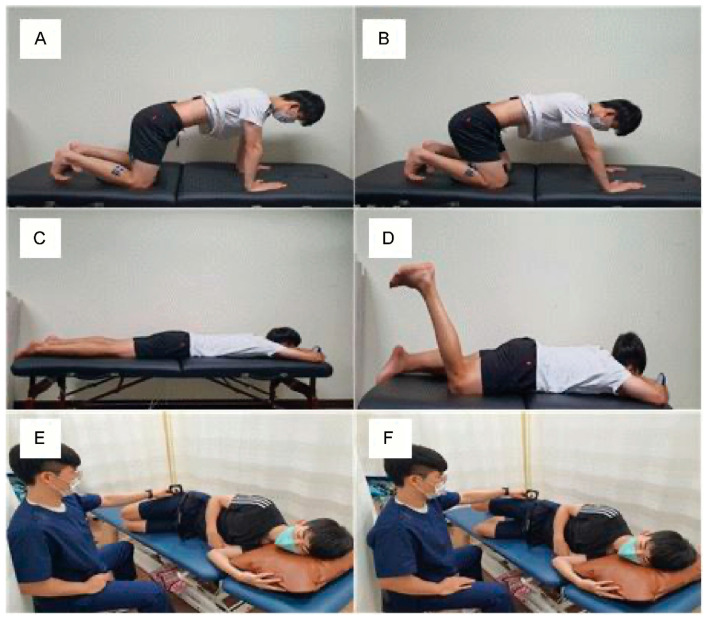
Motor control training. Flexion training: (**A**) start position and (**B**) end position. Extension training: (**C**) start position and (**D**) end position. Rotation training: (**E**) start position and (**F**) end position.

**Figure 6 medicina-59-01271-f006:**
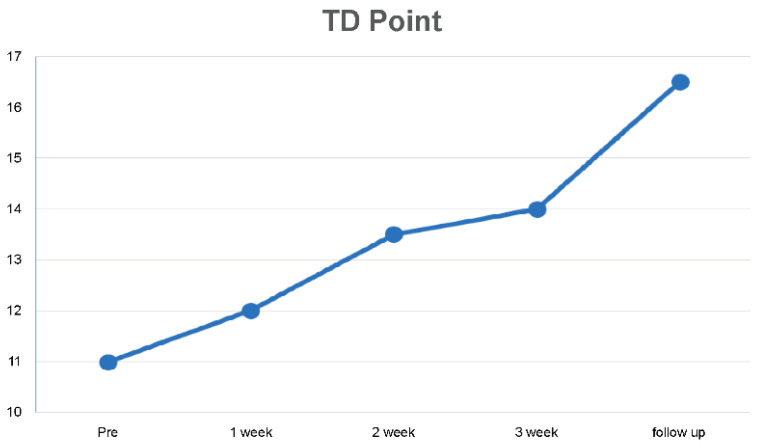
Changes in tactile discrimination point over 3 weeks and during follow-up.

**Figure 7 medicina-59-01271-f007:**
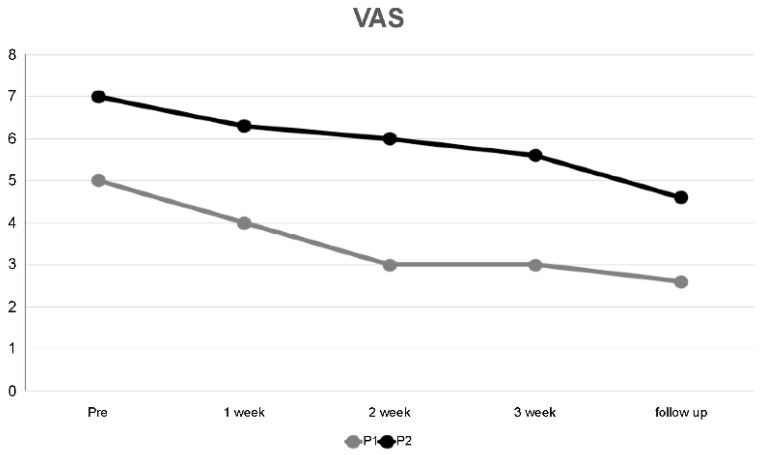
Changes in visual analog scale score for P1 and P2 over 3 weeks and during follow-up.

**Figure 8 medicina-59-01271-f008:**
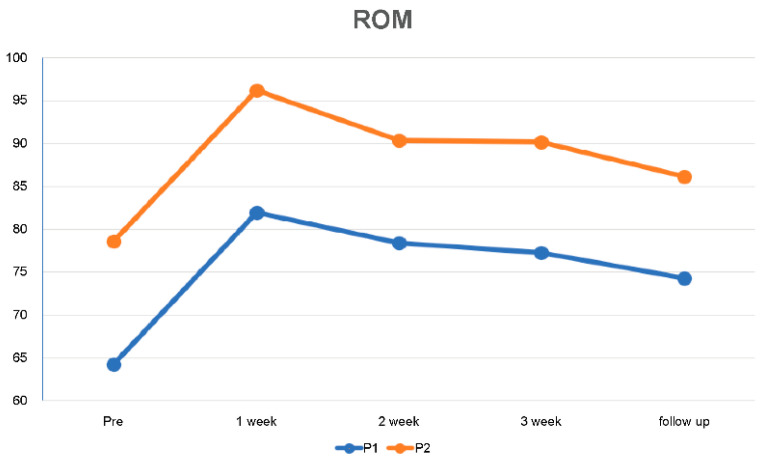
Changes in range of motion at pre-test measurement P1 and P2 over 3 weeks and during follow-up.

**Figure 9 medicina-59-01271-f009:**
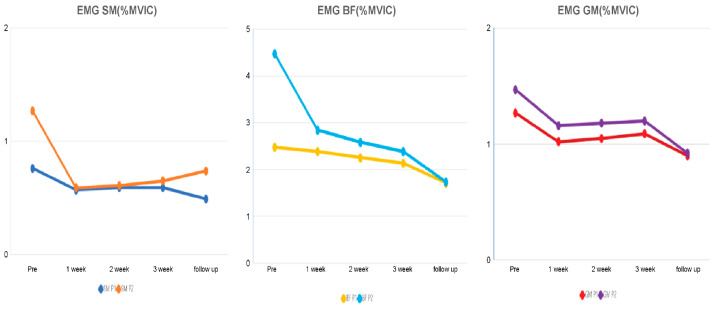
Changes in electromyography activity scores for the semimembranosus, biceps femoris, and gastrocnemius muscles at P1 and P2 over 3 weeks and during follow-up.

**Figure 10 medicina-59-01271-f010:**
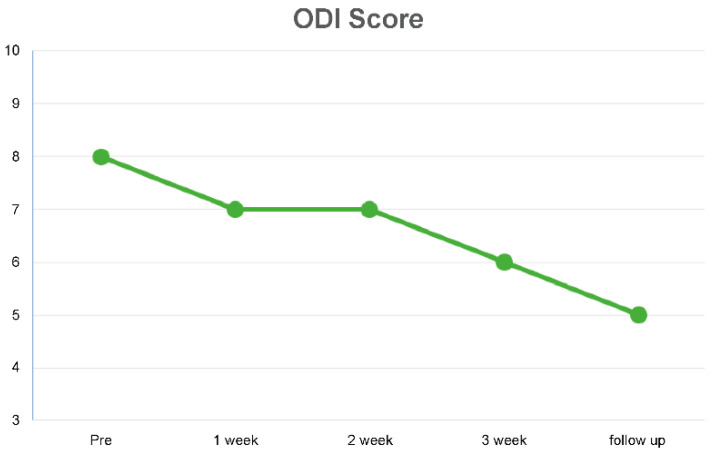
Changes in Oswestry Disability Index score over 3 weeks and during follow-up.

**Table 1 medicina-59-01271-t001:** Patient characteristics.

Variable	Mean ± SD
Age (years)	23
Height (cm)	178
Weight (kg)	67
VAS	5
ODI	8

VAS: visual analog scale; ODI: Oswestry Disability Index.

**Table 2 medicina-59-01271-t002:** Outcomes of TDP in the low back area.

	Pre-Test	1 Week	2 Weeks	3 Weeks	After 2 Weeks
TDP	11	12	13.5	14	16.5

TDP: tactile discrimination point; PRE:

**Table 3 medicina-59-01271-t003:** Outcomes of VAS for ASLR P1 and P2.

		Pre-Test	1 Week	2 Weeks	3 Weeks	After 2 Weeks
VAS	P1	5	4	3	3	2.6
	P2	7	6.3	6	5.6	4.6

VAS: visual analogue scale; ASLR: active straight leg raise; P1: point of onset of pain; P2: point maximum pain.

**Table 4 medicina-59-01271-t004:** Outcomes of ROM in hip joint for ASLR P1 and P2.

	ROM	Pre-Test	1 Week	2 Weeks	3 Weeks	After 2 Weeks
ROM	P1	64.30°	81.93°	78.37°	77.27°	74.27°
	P2	78.57°	96.17°	90.33°	90.20°	86.20°

ROM: range of motion (°); ASLR: active straight leg raise; P1: point of onset of pain; P2: point of maximum pain.

**Table 5 medicina-59-01271-t005:** EMG activity in the ROM of P1 and P2 during active straight leg raise.

	ROM	Pre-Test	1 Week	2 Weeks	3 Weeks	After 2 Weeks
SM (%MVIC)	P1 (64°)	0.76	0.57	0.59	0.59	0.49
	P2 (78°)	1.27	0.59	0.61	0.65	0.74
BF (%MVIC)	P1 (64°)	2.47	2.38	2.25	2.13	1.71
	P2 (78°)	4.47	2.28	2.58	2.38	1.73
GM (%MVIC)	P1 (64°)	1.27	1.02	1.05	1.09	0.90
	P2 (78°)	1.47	1.16	1.18	1.20	0.92

EMG: electromyography; ROM: range of motion (°); P1: point of onset of pain; P2: point of maximum pain; SM: semimembranosus; BF: biceps femoris; GM: gastrocnemius.

**Table 6 medicina-59-01271-t006:** Outcome of ODI score.

	Pre-Test	1 Week	2 Weeks	3 Weeks	After 2 Weeks
ODI score	8	7	7	6	5

ODI: Oswestry Disability Index.

## Data Availability

The data presented in this study are available on request from the corresponding author.

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
