# Peer review of "Effects of Complex Pain Control Programs on Taekwondo Athletes with Recurrent Low Back Pain: A Case Study"

_medicina, 2023, doi:10.3390/medicina59071271_

Round 1
Author Response
Dear. reviewer
We thank you and the reviewers for your thoughtful suggestions and insights.
The manuscript has been rechecked, and the reviewers’ suggestions have made the necessary changes. The responses to all comments have been prepared and attached herewith/given below.
Thank you for your consideration. I look forward to hearing from you.
Sincerely,
Ju-hyeon Jung PT, PhD.

Reviewer 2 Report
This was a case study assessing the effect of a “pain control program” on the development of low back pain in a taekwondo athlete with recurrent low back pain (LBP).
The numerical rating pain scale (NRPS), pain pressure threshold, mechanosensitivity, and Oswestry Disability Index (ODI) scores were measured before and after three weeks intervention. Muscle activation response wa salso assessed through surface EMG assessment of the leg muscles (only as regards the painful side).
The results obtained showed a reduction in Numerical Rating Pain Scale (NRPS) and mechanosensitivity, improvement in range of motion and tactile discrimination abilities, as well as physical function (as assessed by the Owestry Disability Index (ODI). The effect of the improved intervention lasted 2 weeks. Based on the results obtained, authors concluded that application of the proposed pain control programs can be used as an effective intervention in Taekwondo athletes with recurrent LBP.
The topic investigated is of interest, especially considering the target poplation identified (Taekwondo athletes) but the methodological approach used and the conclusions addressed are, in my opinion, not fully supported by the presented data, also because we are talking about a case study. At the end of a case study, suggestions can/could be acceptable, but no conclusion can be addressed.
This reviewer has several majur concerns as regards the methodological approach used, as described in the following lines.
METHODS (lines 128-148): About EMG assessment. Authors assessed EMG through surface bipolar elecrodes from Biceps Femoris and Semitendinoseous muscles, and as they wrote, from the “calf muscles”. This information cannot be enough. Which muscles are authors considering? There are no enough details as regards surface EMG assessment.
Moreover, there is a major methodological issue that has to be carefully considered, and that can be summarized through the following questions:
Why these muscles? Why did authors consider only the leg muscles?
Which is the rationale suggesting this is the best assessment?
Where is the literature background?
Why was the assessment performed only on the painful side and not also the contralateral one?
Why didn’t authors consider the assessment of muscle activation of lumbar muscles (paraspinal) along with abdominal ones?
To this respect, it is quite known that in case of LBP induced by a herniated intervertebral disc, the activation of the muscles of the lower limbs can provide for sure useful information, but when we talk about LBP we cannot ignore the response provided by the muscular activity of the lumbar tract: the authors correctly tested and evaluated the lumbar area response through a rating pain scale and Owestry Disability Index, but they should have also tested the muscle response in this area through EMG assessment. It is widely documented that in the presence of low back pain, the paravertebral musculature provides crucial information in terms of muscle response, in association with the presence of lumbar pain, and this response can also be evaluated through an analysis of the surface EMG signal collected in that area. It has also been demonstrated that a good muscle balance between the anterior (abdominal) and posterior (lumbar) muscles can represent a protective factor against the development of lumbar pain. Therefore, completely ignoring this aspect is, in my opinion, a major limitation in this study.
MINOR COMMENTS
Line 73. Authors wrote: “….He was diagnosed with an L5 hernia of the intervertebral disc…….”. L4-L5 or L5-S1? It should be specified.
Lines 281-282. Authors wrote: …… This phenomenon could also increase the pain in patients to induce involuntary contraction of the muscles in the vicinity, thereby causing pain [Ref 32].” Right. But you assessed leg muscles, not the lumbar tract, that clearly represent the so called “ muscles in the vicinity”.
Lines 342-343. LIMITATIONS OF THE STUDY.
Authors wrote: “….This study had some limitations. First, the lack of a control group prevented a comparison of the effects of the complex intervention programs with those of the single intervention programs.
In my opinion this is not really a limitation. In fact, this is a case study, so of course you cannot compare the results obtained on a control group, it seems a non-sense observation. Authors should indeed be more careful in providing general indications based on the results obtained in a single case study.
Author Response

(The authors gave the same response as above.)

Round 2
Reviewer 2 Report
The manuscript has been modified and improved following the reviewers' suggestions.
To this reviewer's opinion, no additional changes are needed.